# Evaluating systematic targeted universal testing for tuberculosis in primary care clinics of South Africa: A cluster-randomized trial (The TUTT Trial)

Neil A. Martinson[1,2]*, Bareng A. S. Nonyane[3], Leisha P. Genade[1], Rebecca H. Berhanu[4], Pren Naidoo[5], Zameer Brey[6], Anthony Kinghorn[1], Sipho Nyathi[7], Katherine Young[8], Harry Hausler[8], Lucy Connell[9], Keeren Lutchminarain[10,11], Khine Swe Swe-Han[10,11], Helena Vreede[12], Mohamed Said[13,14], Nina von Knorring[15,16], Lawrence H. Moulton[2], Limakatso Lebina[1,17], the TUTT Trial team¶

1 Perinatal HIV Research Unit (PHRU), University of the Witwatersrand, Johannesburg, South Africa, 2 Johns Hopkins University Center for TB Research, Baltimore, Maryland, United States of America, 3 Johns Hopkins Bloomberg School of Public Health, Department of International Health, Baltimore, Maryland, United States of America, 4 Department of Medicine, Division of Infectious Diseases, Vanderbilt University Medical Center, Nashville, Tennessee, United States of America, 5 Public Health Management Consultant, South Africa, Johannesburg, South Africa, 6 Bill and Melinda Gates Foundation, South Africa, Johannesburg, South Africa, 7 AQUITY Innovations, Centurion, South Africa, 8 TB HIV Care, Cape Town, South Africa, 9 Right to Care, Centurion, South Africa, 10 National Health Laboratory Service Department of Microbiology, Inkosi Albert Luthuli Central Hospital, eThekwini, South Africa, 11 University of Kwa Zulu Natal, Durban, South Africa, 12 National Health Laboratory Service, Chemical Pathology, Groote Schuur Hospital, Cape Town, South Africa, 13 National Health Laboratory Service, Microbiology and Academic Division, Tshwane, South Africa, 14 Department of Medical Microbiology, University of Pretoria, Pretoria, South Africa, 15 National Health Laboratory Service, Clinical Microbiology, Johannesburg, South Africa, 16 Division of Clinical Microbiology and Infectious Diseases, University of the Witwatersrand, Johannesburg, South Africa, 17 Africa Health Research Institute, KwaZulu Natal, South Africa

¶ Membership of TUTT Trial team is provided in the Acknowledgements.
* martinson@phru.co.za

**Data Availability Statement:** Data cannot be shared publicly because of local IRB requirements. Data are available for researchers who meet requirements for access to this data. The data underlying the results presented in the study are available from the PHRU Data Centre

## Abstract

### Background

The World Health Organization (WHO) recommends systematic symptom screening for tuberculosis (TB). However, TB prevalence surveys suggest that this strategy does not identify millions of TB patients, globally. Undiagnosed or delayed diagnosis of TB contribute to TB transmission and exacerbate morbidity and mortality. We conducted a cluster-randomized trial of large urban and rural primary healthcare clinics in 3 provinces of South Africa to evaluate whether a novel intervention of targeted universal testing for TB (TUTT) in high-risk groups diagnosed more patients with TB per month compared to current standard of care (SoC) symptom-directed TB testing.

### Methods and findings

Sixty-two clinics were randomized; with initiation of the intervention clinics over 6 months from March 2019. The study was prematurely stopped in March 2020 due to clinics restricting access to patients, and then a week later due to the Coronavirus Disease 2019 (COVID-

(Swanepoelf@phru.co.za), after the local IRB has acknowledged both the planned analysis and there is a fully executed data transfer agreement - a version of which has been pre-approved by the local IRB.

**Funding:** Laboratory tests were funded unconditionally by the Department of Health of South Africa. which had no role in the study design, data collection, analysis, decision to publish or preparation of the manuscript. Research costs were funded by a grant to the Wits Health Consortium (Pty) Ltd by the Bill and Melinda Gates Foundation (BMGF) to NAM Grant#: OPP1191585. Two authors who are employed by BMGF (PN and ZB) reviewed the study protocol but played no role in data collection or analysis. RH receives funding from the NIH which had no involvement in this study (K08AI150352 and T32AI052074).

**Competing interests:** I have read the journal's policy and the authors of this manuscript report one competing interest: NAM's institution receives funding from Pfizer which funds an unrelated observational study. Pfizer had no role in the design or conduct of this trial.

**Abbreviations:** ART, antiretroviral therapy; CDW, Corporate Data Warehouse; COVID-19, Coronavirus Disease 2019; CRP, C-reactive protein; CV, coefficient of variance; DSMB, data safety monitoring board; GP, Gauteng; IRR, incidence rate ratio; KZN, KwaZulu-Natal; LAM, lipoarabinomannan; NHLS, National Health Laboratory Service; PCR, polymerase chain reaction; SARS-CoV-2, Severe Acute Respiratory Syndrome Coronavirus; SoC, standard of care; TB, tuberculosis; TUTT, targeted universal testing for TB; WC, Western Cape; WHO, World Health Organization.

19) national lockdown; by then, we had accrued a similar number of TB diagnoses to that of the power estimates and permanently stopped the trial. In intervention clinics, attendees living with HIV, those self-reporting a recent close contact with TB, or a prior episode of TB were all offered a sputum test for TB, irrespective of whether they reported symptoms of TB. We analyzed data abstracted from the national public sector laboratory database using Poisson regression models and compared the mean number of TB patients diagnosed per clinic per month between the study arms. Intervention clinics diagnosed 6,777 patients with TB, 20.7 patients with TB per clinic month (95% CI 16.7, 24.8) versus 6,750, 18.8 patients with TB per clinic month (95% CI 15.3, 22.2) in control clinics during study months. A direct comparison, adjusting for province and clinic TB case volume strata, did not show a significant difference in the number of TB cases between the 2 arms, incidence rate ratio (IRR) 1.14 (95% CI 0.94, 1.38, $p = 0.46$). However, prespecified difference-in-differences analyses showed that while the rate of TB diagnoses in control clinics decreased over time, intervention clinics had a 17% relative increase in TB patients diagnosed per month compared to the prior year, interaction IRR 1.17 (95% CI 1.14, 1.19, $p < 0.001$). Trial limitations were the premature stop due to COVID-19 lockdowns and the absence of between-arm comparisons of initiation and outcomes of TB treatment in those diagnosed with TB.

## Conclusions

Our trial suggests that the implementation of TUTT in these 3 groups at extreme risk of TB identified more TB patients than SoC and could assist in reducing undiagnosed TB patients in settings of high TB prevalence.

## Trial registration

South African National Clinical Trials Registry DOH-27-092021-4901.

## Author summary

### Why was this study done?

- Tuberculosis (TB) diagnosis is made by laboratory sputum testing usually prompted by the presence of at least 1 symptom of TB (cough, fever or night sweats, loss of weight).

- The World Health Organization promotes screening of entire groups of people at high risk of TB irrespective of symptoms. In South Africa, high-risk groups include people living with HIV, those who report being a close contact of someone with TB, and those who have had a recent episode of TB.

- There is little data on the usefulness of diagnostic assays as screening tests. Our trial assessed if targeted sputum testing of people at high risk without TB symptoms helped to identify undiagnosed TB.

## What did the researchers do and find?

- A total of 62 large primary care clinics in South Africa were randomized to either our intervention—TB testing of sputum of all people in high-risk groups, or our control—to continue symptom directed testing for TB (the standard of care).

- The total number of TB cases diagnosed each month in all clinics was recorded. Comparisons were made between intervention and control clinics.

- In trial intervention clinics, 6,777 people were diagnosed with TB, an average of 20.7 patients per clinic month versus 6,750 in control clinics, an average of 18.8 patients with TB per clinic month.

- After adjusting for clustering, province, and the strata of average number of TB patients diagnosed per month in the last quarter of 2017, intervention clinics diagnosed 14% more patients with TB than control clinics, but this did not reach statistical significance. Secondary analyses, including data from the year prior to the intervention, demonstrated a statistically significant increase in TB diagnoses per month, reported as a 17% relative increase.

## What do these findings mean?

- A strategy targeting high-risk groups for universal testing for TB (TUTT) may help to improve diagnostic rates of TB in areas where prevalence is high.

- Universal testing strategies could be be implemented in low-resource settings provided costs of testing were addressed, and other locally relevant high-risk groups for TB could be targeted.

## Introduction

The World Health Organization's (WHO) 2020 Global Tuberculosis (TB) Report highlighted the gap between the number of new patients diagnosed with TB worldwide (7.1 million) and the estimate of total incident cases (10 million) [1,2], suggesting that a substantial proportion of people with TB disease are missed by health systems. Missed or delayed diagnosis of TB results in additional TB transmission as, in the absence of TB treatment, infectious respiratory droplets continue to be dispersed, and additional morbidity and mortality [3,4]. Although there has been a downward trajectory of total patients diagnosed with TB in South Africa since 2008 [1], the recent 2018 South African TB prevalence survey suggests that 150,000 people with TB were not diagnosed or started on TB treatment in South Africa [5].

Passive case finding, which relies on people with symptoms of TB presenting themselves to the health system, does not identify the majority of missing TB patients [6], and WHO now recommends systematic active case finding directed to those with high risk of TB disease [7]. The most recent WHO guideline on systematic screening for TB includes rapid molecular tests and chest X-rays as primary screening tools in addition to symptom screening to identify those who should be laboratory tested [8]. South African guidelines at the time of the study recommend that if a TB symptom screen identifies at least 1 symptom, this should prompt

collection of a specimen for laboratory TB testing. Symptom screening remains the mainstay of identifying people who require further investigation for TB as it maximizes sensitivity of laboratory testing and reduces laboratory resources being spent on those with a low probability of TB disease [9]. However, several studies report poor sensitivity of symptom screening at the primary care clinic level in people living with HIV, pregnant women, and adults living with HIV receiving antiretroviral therapy (ART) [10–12]. Moreover, TB symptom screening for every clinic attendee together with collection and laboratory testing of a sputum specimen appears to be a difficult target to achieve in South Africa [13].

We did not conduct a formal review of prior research into finding missing patients. There are 3 recent reviews of existing data directed at finding missing TB patients, one of which assessed cost effectiveness; all conclude that in high-risk groups, Xpert testing could be included as a screening tool [14–16].

We posited that there were patients who did not report TB symptoms for some reason, or that they did not have TB symptoms that could be elicited by routine screening, or they did have symptoms but these were not detected or responded to by the health system. Although there are multiple groups at high risk for TB disease [1], there are 3 large groups of adults in South Africa at extreme risk of TB. Firstly, despite being attenuated by ART and TB preventive treatment, the risk of TB disease in people living with HIV remains persistently higher than in their seronegative peers [17]. Secondly, close contacts of a patient with pulmonary TB have a high risk of developing TB disease, particularly in the year following exposure [18,19]. Screening contacts for TB who are attending clinics is likely less costly in identifying additional patients with TB than outreach household and workplace contact tracing. Thirdly, adults with a prior episode of TB are at elevated risk of recurrent TB [20–24]. In South Africa, annual TB incidence was 805 and 738 per 100,000 in 2016 and 2017, respectively [2], and South African TB surveillance data suggest that participants self-reporting prior TB treatment were at least 3 times more likely to have prevalent TB than those who did not [25].

We therefore hypothesized that targeting high-risk groups for universal TB testing, irrespective of whether they report symptoms or not, would identify additional patients with TB. We conducted a cluster-randomized trial of targeted universal testing for TB (TUTT) for clinic attendees in these 3 high-risk groups to ascertain whether more patients were diagnosed with TB every month in those clinics randomized to TUTT than in the control clinics.

## Methods

### Ethics statement

This trial protocol (S1 Information) was approved by the University of the Witwatersrand Human Research Ethics Committee (Medical) (Reference No:180808) and 3 Provincial Research Committees. All participants were ≥18 years and provided their own written informed consent, administered by a study fieldworker. The trial was registered with the South African National Clinical Trials Registry (DOH-27-092021-4901). A data safety monitoring board (DSMB) of 5 individuals was constituted prior to study start to provide trial oversight. The DSMB reviewed the study design prior to study start, met twice during recruitment, and reviewed preliminary comparative analyses after study recruitment was prematurely stopped due to Coronavirus Disease 2019 (COVID-19).

### Study design and setting

A 2-arm, cluster-randomized trial of public sector primary healthcare clinics was implemented in 3 of 9 provinces in South Africa: Gauteng (GP), KwaZulu-Natal (KZN), and Western Cape (WC) (selected because they contribute ≥50% of the annual national TB burden) [26]. The

trial intervention started in some clinics in March 2019, and over 6 months, all intervention clinics were implementing TUTT until COVID-19 lockdowns permanently stopped new recruitment on 20 March 2020. In South Africa, at the time of the study, national guidelines required that all clinic attendees be TB symptom-screened at each visit, and those with at least 1 symptom suggestive of TB should provide a sputum sample for laboratory testing with the Xpert MTB/RIF Ultra (Ultra) (Cepheid, Sunnyvale, CA) polymerase chain reaction (PCR) assay. Virtually, all clinics in South Africa have access to Ultra testing of sputum, free of charge to the patient, and although Ultra testing laboratories are distant from most clinics, results are usually available at the clinic within 2 working days [8].

Clinics were randomized 1:1 either to the control arm where the standard of care (SoC) - symptom-based TB screening sputum testing - continued without change, or to the intervention, which augmented SoC with targeted universal testing for clinic attendees who were in one of the study-defined high-risk groups: living with HIV, self-reported contact of a TB patient within the past year, or diagnosed with TB in past 2 years. Intervention clinics were planned to implement targeted universal testing for TB for a continuous duration of 14 months to account for seasonal variation in clinic attendances and TB diagnoses, and to allow a lead-in month for integration of the intervention into clinic processes. To account for secular trends in TB diagnoses, we collected study outcome data for clinics in both arms using identical processes from a year prior to the first clinic receiving the intervention through to the last intervention clinic completing the intervention phase.

## Eligibility

Trial clinics were identified using data provided by the National Health Laboratory Service's (NHLS) Corporate Data Warehouse (CDW) [27], which includes individual patient results for all laboratory specimens collected by public sector clinics and analyzed in NHLS laboratories. We initially required eligible clinics to diagnose ≥15 individual patients with laboratory-confirmed TB every month based on CDW data from the last quarter of 2017 and the first quarter of 2018. This eligibility threshold was later reduced to ≥10 laboratory-confirmed patients with TB diagnosed per month as there were insufficient clinics in the 3 study provinces diagnosing ≥15 TB patients per month. We excluded clinics in prisons, clinics conducting research that could either interfere with the intervention or confound our outcome, and an outlier clinic in central Johannesburg that diagnosed ≥50 TB patients per month. Additionally, to mitigate potential contamination between clinics, we required eligible clinics be ≥5 kilometers apart. After exclusions, there were 143 potentially eligible clinics diagnosing ≥10 new patients with TB per month and among these, 60 (8 GP, 26 KZN, 26 WC) were randomly selected (Fig 1).

## Randomization and masking

Randomization of selected clinics was stratified by province and by strata of the number of TB patients diagnosed in each clinic per month (10 to 15, 15 to 21, and 21 to 30 patients with laboratory-confirmed TB per month) using CDW data from the last quarter of 2017. Randomization code was written such that for each province and size stratum, clinics were randomly selected to be in either arm with equal probability. The trial statistician initially randomized 8 clinics in GP (a province with relatively few clinics that met eligibility criteria), 26 in KZN, and 26 in the WC (Fig 1). However, implementation was hindered in some due to delays in obtaining local approvals to conduct research; 5 and 7 clinics randomized to control and intervention arms, respectively, were denied approval. Moreover, a clinic randomized to the intervention was destroyed by fire before the start of the intervention, and another a month after the intervention started—data from both are not included in these analyses. Additionally, 4 months

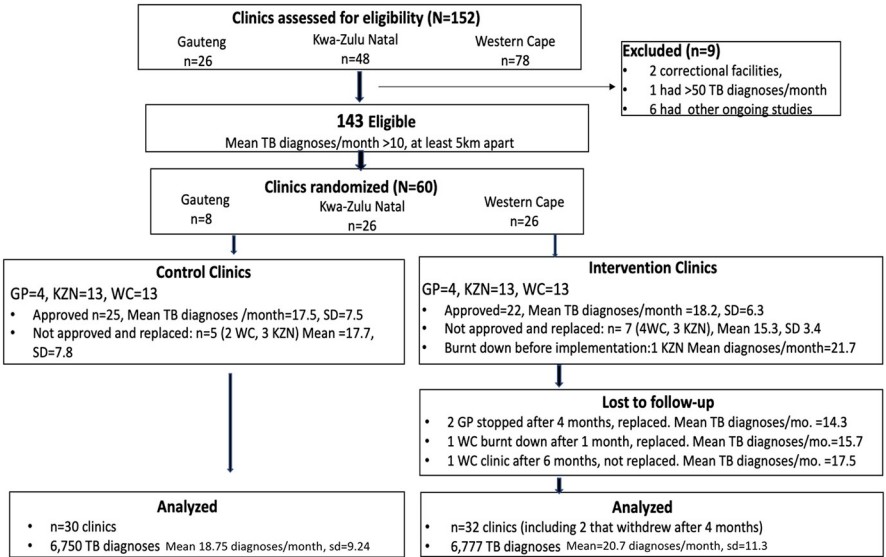

**Fig 1. CONSORT diagram showing clinic selection and TB diagnoses in a cluster-randomized trial of targeted universal testing for clinic attendees at high risk for TB (HIV-infected or recent close contact of a TB patient or recently diagnosed with TB).**

into the intervention, research activities at 2 intervention clinics in GP were halted because another study targeting patients with TB had started, and our trial interfered with the other's eligibility criteria. Clinics were replaced following additional stratified randomization (Table 1). Study investigators responsible for implementation were deliberately masked to outcome data, which were received from the NHLS only after the last patient was recruited. Interim comparative analyses were presented in closed sessions to the DSMB.

## Intervention

A study fieldworker was allocated to be present at each intervention clinic on working days. Fieldworkers received trial-specific training that included approaching and obtaining

**Table 1. Characteristics of randomized clinics by arm stratified by province using actual data or available published data close to the time of initiation of the intervention.**

|  | Gauteng | | KwaZulu-Natal | | Western Cape | | Overall | |
|---|---|---|---|---|---|---|---|---|
|  | Control | Intervention | Control | Intervention | Control | Intervention | Control | Intervention |
| Number of clinics | 4 | 6 | 13 | 13 | 13 | 13 | **30** | **32** |
| Total clinic months over entire duration of study | 48 | 46 | 156 | 150 | 156 | 119 | **360** | **315** |
| Median clinic months (IQR) | 12 | 9 (4,10) | 12 | 12 (11,13) | 12 | 8 (7,12) | **12** | **11 (8,12)** |
| Median number of monthly Xpert Ultra tests prior to study start* | 112.4 | 156.5 | 283.8 | 210.5 | 135.9 | 76.8 | **139.3** | **209.7** |
| Median number of TB patients diagnosed each month prior to study start* | 12 (10,15) | 14 (10,16) | 15 (13,18) | 16 (15,19) | 16 (13,21) | 13 (12,17) | **15 (11,20)** | **16 (13,18)** |
| HIV prevalence in the District where each clinic was located (2017) | 17.6 | 17.6 | 27 | 27 | 12.6 | 12.6 | **19.5** | **19.2** |
| Annual TB incidence (per 100,000) in the District where each clinic was located (2017) | 319 | 322 | 691 | 718 | 695 | 713 | **644** | **644** |

*Including the 2 clinics with 3 months' data (median = 9, IQR 4, 10).

individual written consent from participants, safe collection and labeling of sputum specimens, and the maintenance of patient confidentiality. We used 3 methods to identify potential participants: fieldworkers presented at approximately 2 hourly intervals in intervention clinic waiting rooms during which they introduced the study and its purpose, and requested attendees who thought they were in a high-risk group to privately contact the fieldworker at any time during their clinic visit; they reviewed patient files to assess potential eligibility of participants; and other healthcare workers in intervention clinics were requested to refer potentially eligible patients to fieldworkers for enrolment and sputum collection.

Adults, ≥18 years of age, with at least one of the following were eligible: documented HIV infection in the clinical record irrespective of ART treatment or CD4 count; self-report of a close contact with a person with TB in the workplace or at home or elsewhere in the prior year; or self-report of TB disease—including receipt of TB treatment—in the prior 2 years. We excluded individuals taking TB treatment at the time of the study visit, as well as those who had a trial specimen collected in the previous 6 months. Consented participants had a brief interview, and confirmation of HIV seropositivity was abstracted from medical records, with the most recent CD4 count, if available. Irrespective of the presence of symptoms, every participant had a supervised sputum specimen taken in a standardized manner according to an operating procedure on which fieldworkers rceived training at the beginning and also oversight throughout the trial. Specimens were taken either outside the clinic or in a well-ventilated sputum collection booth. After an oral water rinse, fieldworkers collected ≥3 ml of sputum in a prelabelled specimen container. Participants unable to produce a mucoid sputum specimen immediately were encouraged to make repeated cough efforts and then spit whatever was in their mouth into the specimen container and to repeat this process 3 times. Specimens were couriered by routine transport systems to NHLS mycobacterial culture laboratories (in eThekwini, Cape Town, Tshwane, and Johannesburg) with automated liquid culture capacity.

At each laboratory, after homogenization and decontamination with *N*-acetyl-l-cysteine (NALC), the specimen was centrifuged, and the resulting pellet split in equal parts. One part was subjected to Ultra and the other to the Mycobacterial Growth in Tube (MGIT) automated liquid culture system (Becton Dickinson, Franklin Lakes, NJ). All NHLS TB culture laboratories have quality assurance systems that include analysis of blinded specimen panels and contamination rate review. In intervention clinics, fieldworkers were instructed to follow up any laboratory results that reported *Mycobacterium tuberculosis* detected and to notify the clinic's TB focal point to expedite TB treatment. Ultra-semiquantitative "trace" results, the lowest threshold of detection of *M. tuberculosis* detected by the Ultra [28], were managed according to provincial clinical algorithms: The WC required a positive culture in patients with a prior history of TB treatment, and all required clinical evaluation and repeat laboratory testing if clinically warranted. Additionally, the study team created an advisory panel of 3 infectious disease physicians who clinic staff could consult for advice. Clinic attendees at intervention clinics who were not enrolled in the trial received usual TB symptom screening with specimen collection, as appropriate, from routine health workers. These specimens were delivered by NHLS courier to the nearest NHLS laboratory to be tested using Ultra.

In control clinics, no changes in TB symptom screening or collection of specimens for TB testing occurred. Patients attending these clinics continued to have specimens collected if they reported TB symptoms. All control clinics' sputum specimens were delivered by routine courier to the nearest NHLS laboratory to be tested using Ultra. TB screening and sample collection processes were observed and evaluated at least twice at each control clinic over the duration of the intervention phase. Clinics that scored poorly on 2 occasions on a standardized evaluation by study staff were brought to the attention of the Provincial TB Manager by the study team.

## Data sources and analyses

The primary outcome was the number of unique patient TB diagnoses reported in the NHLS's CDW database per clinic per month. After deduplicating results, TB test results obtained from CDW spanning February 2018 through March 2020 were included. For the purpose of this trial, we defined a TB patient as: unique individuals with a laboratory result that included *M. tuberculosis* complex detected, either by Ultra and/or liquid mycobacterial culture in all study clinics per month over the study duration; in all study clinics, patients diagnosed with TB for a second time within 6 months of their first positive test were counted once. In intervention clinics, this count included participants diagnosed with laboratory-confirmed TB as part of the intervention and through routine clinic processes for patients who were not eligible for the intervention.

We initially planned a confirmatory second data source for the primary outcome using clinic-maintained TB registers from which study personnel abstracted the total number of TB cases diagnosed by that clinic each month. However, during the period of study implementation, clinic-based TB registers were replaced with an electronic system, Tier.net [29]. It was apparent that clinic TB registers were not standardized; some did not differentiate between patients diagnosed in the clinic and those initially diagnosed elsewhere and referred to the clinic, or between laboratory and clinically diagnosed patients. We report these analyses in sensitivity analyses.

## Sample size calculations

We based sample size on being able to determine at least a 25% difference in number of TB patients diagnosed per clinic per month between intervention arm clinics and SoC arm clinics, assuming a full calendar year of the implementation phase. We wanted to obtain a full calendar year of data from each clinic to account for seasonal changes. To do this, we originally planned to collect 14 months of intervention and control clinic data from each clinic to account for a lead-in month during which the fieldworker was learning to optimize the intervention, and to account for the possibility of a month of clinic closure (due to strikes or unrest; both are frequent in South Africa). Data provided by CDW from the first quarter of 2018 indicated that clinics diagnosed a median of 16 (range 9,66) cases per month (30-day period) across the 3 study provinces. The observed between-facility coefficient of variance (CV) was 0.34, which reduced to 0.1 after stratifying facilities into small, medium, and high case rates. We expected that size and provincial stratification in our study would also reduce the CV because of the variability in 3 provinces' TB burden. We thus conducted power estimates for a CV of 0.1 as well as for a more conservative estimate of 0.24. With a CV of 0.1, 60 clinics (30 per arm) would provide more than 99% power to detect a difference of 25% in the number of TB cases diagnosed per month between the 2 study arms after 6 or 14 months of implementation in an analysis that adjusts for these stratification variables. A CV of 0.24 would provide 91% power after 6, and 93% after 14 months' of observation per clinic. We estimated that, with a coefficient of variation of 0.24 over a period of 14 months' observations per clinic, 60 clinics (30 per arm) would provide 93% power to detect an increase of 25% in the number of TB cases diagnosed per month in the intervention compared to control clinics.

## Statistical analyses

**Primary analysis.**   We report results for the intervention period, comparing the intervention clinics with control clinics. For all analyses, we assumed that those with an Ultra trace result were negative for *M. tuberculosis* complex to prevent biasing trial results toward the intervention [30]. In intervention arm clinics, the primary outcome was contributed to both

by the sputum collections of the intervention and by routine TB diagnoses of the clinic; in the control clinics, this reflected only routine symptom-directed diagnostic processes. Analyses were prespecified in the approved protocol and were further described in a study-specific statistical analysis plan finalized in October 2019. After the last participant was recruited, we obtained CDW TB diagnoses data from 1 year prior to starting the intervention through the last month of the intervention (S1 and S2 Figs).

The primary analysis compared the mean number of TB patients diagnosed per month per clinic between study arms during the intervention period only, using both Ultra and liquid culture results. Unadjusted analyses used the *t* test on cluster-level TB case rates (i.e., TB diagnoses/clinic's observation months), and a robust check was conducted by running the *t* test on the log of the case rates. Furthermore, Poisson regression models with robust variance or a Pearson residual overdispersion parameter to adjust for extravariability due to clustering were used to compare the outcomes between study arms adjusting for the clinic stratification variables used for randomization (province, and the strata of average number of TB patients diagnosed per month in the last quarter of 2017), with the offset as the $\log_{10}$ of the total number of months the clinic participated in the study. For the primary analysis comparing intervention clinics directly to control clinics, we used all months in 32 intervention clinics while the intervention was implemented, and for all 30 control clinics, 12 months of CDW data from March 2019 to February 2020 were included.

In a sensitivity analysis, we applied Poisson regression—excluding each intervention arm clinic's first month of intervention data to account for a possible initial learning period and a regression excluding the 2 GP clinics that had only 4 months of activity before being replaced. A sensitivity test was applied to this primary analysis but using clinic-based TB count data. Finally, although not protocol specified, we report the primary outcome modeling the results as though the Ultra were the only assay used, as it is unlikely that laboratory capacity to conduct large numbers of mycobacterial cultures would be feasible. Additionally, we also report results assuming trace results were included as positive Ultra result.

To assess the impact of the study fieldworker, we also report a dose–response test, using the Poisson regression, within the intervention clinics to evaluate if the number of days that study fieldworkers were present in the clinic per study month was associated with the number of TB cases diagnosed.

**Secondary analyses.** A protocol-specified difference-in-differences analysis was conducted, comparing the number of TB cases between individual clinics by study arm during the months of the intervention and during the corresponding calendar months for each clinic from the prior year. Thus, for control clinics, a total of 24 months' data were included per clinic, and for intervention arm clinics, this varied from 8 to 26 months per clinic. We used a Poisson regression model including study arm, period (study and prestudy), and their interaction, as well as province and size strata used for randomization. A final combined analysis including an arm-by-period interaction term to ascertain the overall benefit of the intervention strategy taking into account year-on-year changes. We conducted stratified analyses by size, province, gender, and type of test (Ultra versus liquid mycobacterial culture). All data analyses used Stata 16 [31].

## Results

Eleven intervention arm clinics initiated universal testing in March 2019; over the following 6 months, all intervention clinics were initiated, with the last clinic initiating the intervention in October 2019. In the latter part of March 2020, as Severe Acute Respiratory Syndrome Coronavirus 2 (SARS-CoV-2) diagnoses in South Africa increased, patient access to clinics was

restricted, and those with respiratory symptoms were not tested for TB [32]. The study was stopped on 20 March 2020, a week prior to start of a national lockdown. Once it was apparent that lockdown would persist, the study team permanently stopped the study, a decision ratified by the TUTT trial DSMB, which determined that sufficient data were available to draw inferences, after adjusting for each clinic's implementation months, considering that a total of 13,527 TB diagnoses had been made in the trial period, comparable to the 13,662 diagnoses in the corresponding prior period.

Using data extracted from the CDW for all 62 trial clinics' study months, from March 2018 through to March 2020, a total of 295,801 TB laboratory tests. Of these, 222,905 (9.6% positive for *M. tuberculosis*) were Ultra and 66,746 (10.5% positive for *M. tuberculosis*) were liquid mycobacterial culture). Further stratification by test type and positivity rates are in Table 2.

A detailed description of the participants who were consented and tested in intervention clinics, including the proportion who were positive on each or both TB tests, stratified by the high-risk group/s they were in, is described elsewhere [30]. In brief, of 32,891 consented intervention clinic attendees, 30,513 had at least 1 sputum TB laboratory assay result (S1 Table). Their median age was 37 years (IQR 30 to 46), and 11,553 (38%) were men. Most (71%) were

**Table 2. Mycobacterial laboratory assays taken at study clinics, stratified by assay used, sex and age of person tested, with proportion and rates positive for *M. tuberculosis* and presence of rifampicin resistance; by arm and by period.**

| | Pretrial Period | | | | | | Trial Period | | | | | |
|---|---|---|---|---|---|---|---|---|---|---|---|---|
| | Intervention Clinics | | | Control Clinics | | | Intervention Clinics | | | Control Clinics | | |
| | Total Tests | Positive for *M tb* [a], for rif [b] resistance (%) | Rate* #positive tests per clinic month | Total Tests | Positive for *M tb* [a] for rif [b] resistance (%) | Rate* #positive per clinic month | Total Tests | Positive for *M tb*[a] for rif [b] resistance (%) | Rate* #positive per clinic month | Total Tests | Positive [a] for *M tb* [b] for rif resistance (%) | Rate* #positive per clinic month |
| Xpert Ultra (Trace excluded) | 47,660 | [a] 4,911 (11.5) | 15.59 | 58,057 | [a] 5,849 (11.2) | 16.25 | 67,384 | [a] 5,315 (8.6) | 16.87 | 55,954 | [a] 5,286 (10.5) | 14.68 |
| Xpert Ultra Trace result | | [a] 300 (0.7) | 0.95 | | [a] 289 (0.6) | 0.80 | | [a] 1,282 (2.10) | 4.07 | | [a] 861 (1.7) | 2.39 |
| Xpert Ultra Rifampin resistance | | [b] 223 (0.5) | 0.71 | | [b] 260 (0.5) | 0.72 | | [b] 279 0.5 | 0.89 | | [b] 203 (0.4) | 0.56 |
| Liquid mycobacterial culture | 10,323 | [a] 1,503 (17.1) | 4.77 | 10,132 | 1,612 (19.0) | 4.48 | 34,402 | 2,314 (7.3) | 7.35 | 11,889 | 1,567 (15.2) | 4.35 |
| Liquid culture rifampin resistance | | [b] 224 (2.5) | 0.71 | | [b] 292 (3.4) | 0.81 | | [b] 288 (0.9) | 0.91 | | [b] 254 (2.5) | 0.71 |
| [‡]Men | 25,536 | 3,836 (15.0) | 12.18 | 29,737 | 4,320 (14.5) | 12.00 | 30,228 | 4,030 (13.3) | 12.79 | 28,399 | 4,109 (14.5) | 11.41 |
| [‡]Women | 24,952 | 2,292 (9.2) | 7.28 | 32,396 | 2,814 (8.7) | 7.82 | 39,228 | 2,524 (6.4) | 8.01 | 33,474 | 2,481 (7.4) | 6.89 |
| [‡]Unknown sex | 2,036 | 202 (9.9) | 0.64 | 2,036 | 198 (9.7) | 0.55 | 2,645 | 223 (8.4) | 0.71 | 1,123 | 160 (14.2) | 0.44 |
| [‡]0–14 years | 1,763 | 29 (1.6) | 0.1 | 1,134 | 38 (3.4) | 0.1 | 1,324 | 35 (2.6) | 0.1 | 1,199 | 41 (3.4) | 0.1 |
| [‡]15–45 years | 22,770 | 3,148 (13.8) | 9.9 | 28,969 | 3,624 (12.5) | 10.1 | 30,711 | 3,287 (10.7) | 10.5 | 27,717 | 3,303 (11.9) | 9.2 |
| [‡]>45 years | 26,430 | 3,069 (11.6) | 9.7 | 32,281 | 3,520 (10.9) | 9.8 | 39,553 | 3,400 (8.6) | 10.6 | 33,105 | 3,293 (9.9) | 9.3 |
| *[‡]Missing age | 1,561 | 84 (5.4) | 0.3 | 1785 | 150 (8.4) | 0.4 | 1,219 | 55 (4.5) | 0.3 | 1,975 | 113 (5.7) | 0.1 |

Intervention clinics' number of tests in the pretrial period are lower than the number of tests conducted in control clinics because we counted months matched to those of trial intervention period as the comparison.

*Ages >99 years were assumed to be missing.

[‡]Including either Xpert MTB/RIF Ultra or liquid culture assay results.

living with HIV, 41% reported being in close contact with a TB patient, and 5% reported had been diagnosed with TB in the past 2 years. Of those who were tested by the study, 5.1% were positive for *M. tuberculosis* when Ultra trace results were excluded, but this increased to 7.6% when Ultra trace results were considered positive; a caveat is that merely 10% of these Ultra trace results were culture positive. In these participants recruited in intervention clinics, 55% of those with a positive laboratory result did not report TB symptoms on a symptom screen.

Overall, for all intervention clinics during the intervention months, 101,786 specimens were analyzed in the public health laboratory service—this number includes specimens of the intervention participants—compared to 57,983 in the same calendar months a year prior to the intervention (Table 2). In control clinics, 67,843 tests were analyzed during the 12-month intervention period and 68,189 in the same month a year prior. Based on the pretrial period data (Table 2), the assignment of clinics to the 2 study arms was balanced with respect to the proportion of overall Xpert positive tests per clinic month (15.6% versus 16.3%), and positive tests among men (12.2% versus 12.0%)) and women (7.3 versus 7.8) and across age groups.

The 32 intervention clinics contributed a cumulative total of 315 clinic months of intervention time; the median number of months each clinic received the intervention was 11 (IQR 8, 12). CDW data restricted to the months while universal testing was implemented in intervention clinics showed that 6,777 individuals were diagnosed with TB. This included both TB patients diagnosed by the intervention and patients diagnosed through routine symptom-directed TB testing at the clinic, representing a monthly TB case rate of 20.7 cases per clinic (95% CI 16.7, 24.8) (S1 and S2 Figs). Men were 60% (4,030/6,777) of all TB diagnoses in intervention clinics. In matched clinic months from the year prior to the intervention, 6,330 patients were diagnosed with TB in intervention clinics—of whom 4,109 (61%) were men—for monthly case rate of 19.3 (95% CI 15.9, 22.8).

The 30 control clinics contributed 12 study months of CDW data for each clinic for a total of 360 months of data both in the intervention period and in the preintervention period. During this time, 6,750 patients were diagnosed with TB for monthly case rate 18.8 (95% CI 15.3, 22.2) in the year of the intervention, compared to 7,332 patients diagnosed with TB for a monthly case rate of 20.4 (95% CI 16.9, 23.7) in the year preceding the intervention.

Unadjusted counts of positive diagnoses from the intervention period and the year prior to the intervention period, stratified by multiple subgroups, suggest increases in TB diagnoses per month in intervention clinics compared with the prior period for all strata. In the control arm, fewer patients were diagnosed with TB per clinic per month in the intervention period compared to the year prior across these strata.

## Primary comparison: Contemporaneous TUTT versus control clinics

The unadjusted intention to treat between-arm difference in TB diagnoses by either or both Ultra and liquid culture per month was −1.96 (95% CI −7.22, 3.30). The incidence rate ratio (IRR), comparing the 2 arms, using Pearson regression to adjust for province and stratification by the number of TB patients diagnosed, was 1.14 (95% CI 0.94, 1.38, $p = 0.46$). In our study, the unadjusted CV was 0.45 overall (0.46 for the control arm and 0.45 for the TUTT arm), while adjustment for province and clinic size reduced this to 0.29. When we repeated the primary analysis of the laboratory data, including all Ultra trace results from both arms (assuming they were *M. tuberculosis)*, the adjusted IRR for both mycobacterial culture and Ultra results was 1.19 (95% CI 0.99, 1.44, $p = 0.05$). In a second model removing the culture results from both arms and assuming trace as negative, the adjusted IRR was 1.14 (95% CI 0.96, 1.35, $p = 0.13$), but the inclusion of Ultra trace results to Ultra *M. tuberculosis* detected resulted in an increased adjusted IRR of 1.22 (95% CI 1.03, 1.44, $p = 0.02$).

### Secondary analysis: Using data from clinic-maintained registers

Using data from clinic-maintained registers, 7,323 individuals were diagnosed with TB in the TUTT arm at a rate of 22.16 (95% CI 17.24, 27.08) per clinic month, and 7,528 were diagnosed in the control arm at a rate of 20.63 (95% CI 15.92, 25.33). The IRR from the Pearson regression, adjusting for size and provincial stratification, was 1.09 (95% CI 0.87,1.39, $p$ = 0.43).

### Secondary analysis: Difference-in-differences comparison

Clinic-specific year-on-year mean differences in monthly TB case rates for each calendar month varied more in intervention clinics than in control clinics (Fig 2). Nine of 30 clinics in the control arm showed an improvement in TB case rates (median 1.3 [min. 0.25 and max 3.25]). For the intervention facilities, we investigated 2 outlying facilities with a change of −8.9 and −13.2. The data show similar patterns over calender months in the pretrial compared to the trial period except that the latter period values are lower. Each of these outlier clinics had 10 months of clinic activity, with a median of 19.5 (min 11, max 23) days a month when the study fieldworker was available to ensure implementation at the clinic. We were unable to identify a reason for the reduction in TB diagnoses at these 2 clinics.

A difference-in-differences regression model, which included a study-arm by study-period interaction term, indicated that in intervention clinics, there was an *overall increase in monthly TB diagnoses* of approximately 7% with an IRR of 1.07 (95% CI 1.05, 1.08, $p$ < 0.001). In contrast, in control clinics, there was an 8% reduction with an IRR 0.92 (95% CI 0.91, 0.93, $p$ < 0.001) in TB patients diagnosed per month, representing approximately 1.5 fewer TB diagnoses per month. A combined difference-in-differences regression analysis, including an arm-by-period (preintervention and intervention periods) interaction term showed that intervention clinics had a 17% relative increase in patients diagnosed with TB, with an interaction IRR of 1.17 (95% CI 1.14, 1.19, p < 0.001) in the intervention phase compared to the control clinics, representing an increase of approximately 2 additional TB patients diagnosed per clinic per month in intervention clinics. The difference-in-differences analysis was repeated, removing the first month of the intervention and the same calendar month from the year prior to

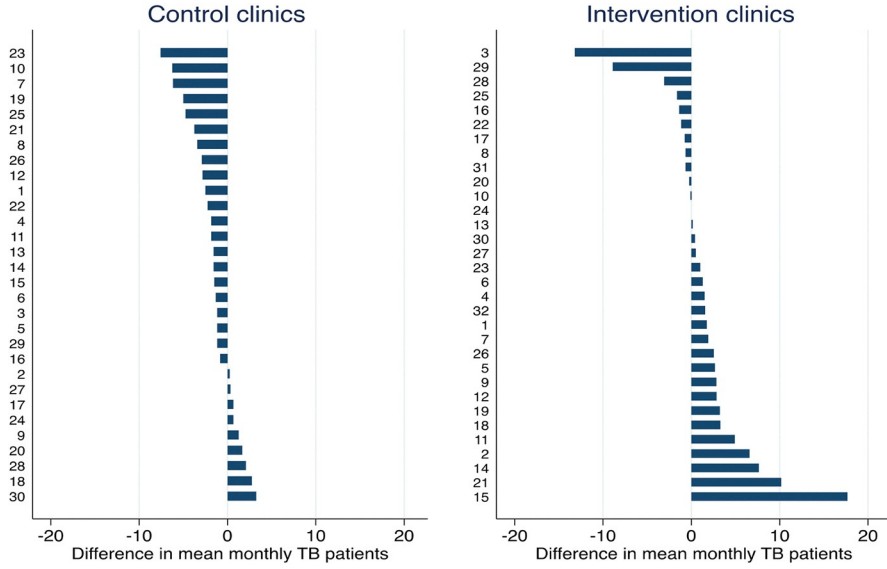

**Fig 2. Ranked unadjusted absolute change in mean monthly TB diagnoses per clinic in the trial months compared to corresponding months in the pretrial calendar year.**

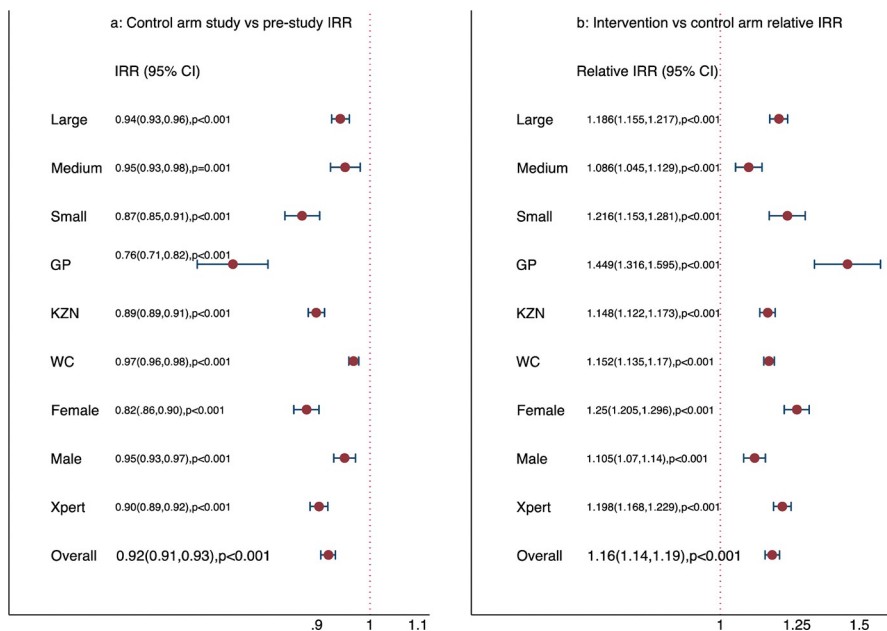

**Fig 3. Results of the difference-in-differences analysis, adjusted for province and size-of-clinic strata, other important variables, and overall; the models included the arm and arm-b-aperiod interaction term.** (a) Change in TB cases diagnosed in the nonintervention clinics from prestudy period. (b) Relative change in TB cases diagnosed in TUTT clinics from the preintervention period to the intervention period. **Footnote**: CI, confidence interval; IRR, incidence rate ratio. Subanalyses are as follows: Large, medium, and small indicates clinic strata defined by the prestudy number of people diagnosed with TB per month, namely 10–15, 15–21, and 21–30, respectively; GP, Gauteng Province; KZN, Kwa Zulu Natal Province; WC, Western Cape Province; Xpert refers to diagnosis methods; the dotted vertical line is the line of no effect (IRR = 1).

account for a learning month, and results remained the same. Stratified differences-in-differences regressions by province, size stratum, gender, and test type showed a significant decline in TB patients diagnosed in control clinics and a relative increase in intervention arm clinics (Fig 3A and 3B).

## Dose–response test in the intervention arm

Study fieldworker absenteeism or clinic/community factors resulting in less half the usual monthly working days in the intervention clinics was experienced in 2 clinic months representing 0.63% of total intervention clinic months. The median number of clinic operation days with the study fieldworkers present at their clinics was 20 (IQR 18, 21). There was a small statistically significant increase in the number of TB cases diagnosed per month, for each additional day that the fieldworkers were present at the clinic with an IRR of 1.03 (95% CI 1.02, 1.03, $p < 0.001$) after adjusting for province and clinic size.

## Discussion

We did not meet our stated goal of diagnosing an additional 25% more adults with laboratory-confirmed pulmonary TB by augmenting symptom-directed TB testing in clinics with systematic universal TB testing of individuals at high risk of TB disease. The primary contemporaneous between-arm comparison was a nonsignificant 14% increase in TB diagnoses in intervention clinics. However, we demonstrated that taking into account the secular decline in TB diagnoses, intervention clinics had a relative year-on-year increase in TB diagnoses per month of 17%.

Novel approaches to diagnose additional patients with TB include augmenting symptom screening with chest X-rays subjected to automated diagnostic algorithms [33] or adding C-reactive protein (CRP) assessment [34,35]. Additionally, we had originally considered including urinary point of care lipoarabinomannan (LAM) assays for people living with HIV whose CD4 count is low [36], but differential testing was considered too complex to be implemented on a large scale. With time, these strategies are becoming more accessible and less expensive. However, routine sputum testing of high-risk groups can be implemented immediately, using healthcare, logistics, and laboratory infrastructure already in place in settings like South Africa. Obtaining specimens from participants who reported no symptoms was feasible and in another manuscript we describe yield of TB testing in the intervention arm, by risk group, and by the presence or absence of TB symptoms [30].

More research is required to assess the cost-effectiveness of this strategy and, importantly, whether it would be appropriate to implement in less-resourced settings than South Africa. The TUTT approach will clearly be more costly and requires higher numbers to be tested (NNT) to identify each additional TB patient when compared to existing symptom-directed TB testing. This study highlights the urgent need for rapid, affordable, and accurate TB screening (and testing) assays that could be applied universally in high-risk populations or, alternatively, other less expensive strategies to reduce costs, such as pooling multiple specimens [37]. During the COVID-19 pandemic, there has been a rapid and sustained reduction in the number of sputum specimens submitted for TB testing [38,39] in South Africa and elsewhere. TUTT strategies could be used to counter this decline in TB diagnoses and has been already explored for other high-risk groups at the level of the household for close contacts of adult and child index TB patients [40–42], pregnant women living with HIV [12], and in people living with HIV starting ART [10,11]. Clearly in other settings, alternative high-risk groups could be tested.

Strengths of our study include its implementation in a large sample of randomly selected public sector clinics in 3 different provinces of South Africa and its use of the public sector laboratory courier and analytical systems already in place to analyze study specimens. The results, therefore, represent what would be achieved if this intervention were scaled up in South Africa.

A key limitation of this study was trial interruption due to the COVID-19 pandemic. Despite stopping the study early, we do not believe that COVID-19 substantively impacted the results we report here, primarily because the number of TB diagnoses in the study period was 13,527, a rate of 20.5 per clinic month, comparable to the 13,440 TB diagnoses (60 clinics over 14 months at an average rate of 16 diagnoses per clinic month) anticipated in our power estimate. Moreover, the final day of recruitment to the study was a week prior to the introduction of lockdown when clinic procedures were starting to be amended in response to concerns about COVID-19. A cumulative total of only 1,278 patients with laboratory-confirmed SARS-CoV-2 had been diagnosed in South Africa by then [38]. Obstacles to implementation led to variations in the number of clinic months of the intervention, which increased between-clinic outcome variability. The trial intervention could be construed to be more than just TUTT as study-hired fieldworkers were placed at intervention clinics to identify and consent high-risk individuals and to collect specimens; moreover, study sputum specimens were not processed in the same manner as routinely collected specimens. Our data are not generalisable to children younger than 18 for whom alternative testing strategies would be required, nor to settings with lower HIV prevalence than South Africa. Finally, we did not assess the effect of the intervention on TB treatment initiation rates, treatment outcomes, or adverse events related to trial procedures or initiation of TB treatment.

In this trial of targeted universal testing for TB, clinics randomized to the intervention diagnosed substantially more individuals by sputum testing than did routine symptom-directed TB testing and indicates that a TUTT strategy could make a contribution to identify "missing" patients with TB in South Africa. The key questions that remain are whether this approach provides a morbidity, mortality, and transmission benefit, and whether it is possible for TUTT to be implemented in a sustainable way, especially in settings with fewer resources or lower HIV prevalence than South Africa.

## Supporting information

**S1 Checklist. Extension for cluster trials.**
(DOCX)

**S1 Table. Characteristics and yield of sputum TB testing in participants recruited in intervention clinics in the cluster randomized trial.**
(DOCX)

**S1 Fig. Average number of patients with TB per clinic, per month in clinics randomized to standard of care (SoC).**
(TIF)

**S2 Fig. Average number of patients with TB per clinic, per month in clinics randomized to targeted universal TB testing (TUTT) intervention.**
(TIF)

**S1 Information. IRB-approved TUTT protocol version 4.0_10 March 2020_.**
(PDF)

## Acknowledgments

Participants who consented to be in the study, the multiple clinics, and their staff that hosted the study. The DSMB (Drs R Rudolfo, P Naidoo, M Zungu, B Girdler-Brown) for considered responses and suggestions to our plans and to interim and final analyses. The study's Diagnostic Advisory Committee (Drs R Berhanu, and E Variava) who provided clinical guidance. The National Health Laboratory Service (NHLS) for scaling up testing and for providing access to its CDW TB data. The South African National Priorities Programme (Drs L Scott, P da Silva, and Mr Gabriel Eisenberg and Mr Paul Ajayi) for retrieving individual clinic data before we started the trial that was used to estimate sample sizes and to identify study clinics.

The TUTT Trial Team: Jacqueline Ngozo, Refilwe Mokgetla, James Kruger, Minja Milovanovic, Floris Swanepoel, Lucia Maloma, Phindiswa Tshobonga, Juanita Chewpersad, Aphiwe Dumezweni, Thembisile Majola, Nhlanhla Mhlongo, Netricia Kooverjee, Debbie Myburgh, Thobeka Lebenya, Dr Bridget Ikhalafeng, Dr. Elizabeth Ohaju, Dr Ronel Kellerman, Lettah Mametse, Peter Silwimba, Dr Elizabeth Lutge, Josh-Lee Kroukamp, Natacha Berkowitz, Sabela Petros, and Judy Caldwell-Taylor.

## Author Contributions

**Conceptualization:** Neil A. Martinson, Limakatso Lebina.

**Formal analysis:** Bareng A. S. Nonyane, Rebecca H. Berhanu, Lawrence H. Moulton.

**Funding acquisition:** Neil A. Martinson, Pren Naidoo, Zameer Brey, Limakatso Lebina.

**Investigation:** Harry Hausler, Lucy Connell, Keeren Lutchminarain, Khine Swe Swe-Han, Helena Vreede, Mohamed Said, Nina von Knorring, Limakatso Lebina.

**Methodology:** Neil A. Martinson, Katherine Young, Harry Hausler, Lucy Connell, Helena Vreede, Mohamed Said, Limakatso Lebina.

**Project administration:** Neil A. Martinson, Leisha P. Genade, Sipho Nyathi, Katherine Young, Lucy Connell, Limakatso Lebina.

**Resources:** Sipho Nyathi, Katherine Young, Lucy Connell, Keeren Lutchminarain, Helena Vreede, Nina von Knorring.

**Supervision:** Neil A. Martinson, Leisha P. Genade, Pren Naidoo, Zameer Brey, Harry Hausler, Limakatso Lebina.

**Validation:** Lawrence H. Moulton.

**Visualization:** Anthony Kinghorn, Harry Hausler, Lawrence H. Moulton.

**Writing – original draft:** Neil A. Martinson.

**Writing – review & editing:** Neil A. Martinson, Rebecca H. Berhanu, Anthony Kinghorn, Sipho Nyathi, Katherine Young, Lucy Connell, Keeren Lutchminarain, Helena Vreede, Mohamed Said, Nina von Knorring, Lawrence H. Moulton, Limakatso Lebina.

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
