## [Editor Report · Decision Letter 0]

7 Oct 2022

Dear Dr Martinson, 

Thank you for submitting your manuscript entitled "A Cluster Randomized Trial of Systematic Targeted Universal Testing for Tuberculosis in Primary Care Clinics of South Africa" for consideration by PLOS Medicine.

Your manuscript has now been evaluated by the PLOS Medicine editorial staff as well as by an academic editor with relevant expertise and I am writing to let you know that we would like to send your submission out for external peer review.

Please re-submit your manuscript within two working days, i.e. by Oct 11 2022 11:59PM.

Kind regards,

Philippa Dodd, MBBS MRCP PhD

Senior Editor

PLOS Medicine

---

## [Decision Letter · Decision Letter 1]

14 Dec 2022

Dear Dr. Martinson,

Thank you very much for submitting your manuscript "A Cluster Randomized Trial of Systematic Targeted Universal Testing for Tuberculosis in Primary Care Clinics of South Africa" (PMEDICINE-D-22-03274R1) for consideration at PLOS Medicine. 

[LINK]

In light of these reviews, I am afraid that we will not be able to accept the manuscript for publication in the journal in its current form, but we would like to consider a revised version that addresses the reviewers' and editors' comments. Obviously we cannot make any decision about publication until we have seen the revised manuscript and your response, and we plan to seek re-review by one or more of the reviewers. 

We expect to receive your revised manuscript by Jan 04 2023 11:59PM. Please email us (plosmedicine@plos.org) if you have any questions or concerns.

We look forward to receiving your revised manuscript. 

Sincerely,

Philippa Dodd, MBBS MRCP PhD

PLOS Medicine

plosmedicine.org

GENERAL

Please respond to all editor and reviewer comments detailed below, in full.

The reviewers agree the study is important and well-conducted, but there are a number of clarifications which are required which the editorial team are in agreement with. Please see below.

Thank you for including a CONSORT flow diagram. Please complete the CONSORT checklist and ensure that all components of CONSORT are present in the manuscript, including [how randomization was performed, allocation concealment, blinding of intervention, definition of lost to follow-up, power statement]. When completing the checklist, please adapt to use section and paragraph numbers, rather than page numbers (which often change at publication).

COMMENTS FROM THE ACADEMIC EDITOR

A major revision makes sense to me at this time. No major issues. The article seems well written and the discussion brought up the key issues that come to mind.

CONFLICT OF INTEREST STATEMENT

Thank you for including a statement. Please clarify whether Pfizer contributed specifically to this study and if so, please describe what role they had in the study design, if any (your current funding statement would suggest none).

FUNDING STATEMENT

Thank you for including a funding statement. Please update, as necessary in line with above.

DATA AVAILABILITY STATEMENT

Thank you for including a Data Availability Statement (DAS) which requires revision. For each data source used in your study: 

ABSTRACT

Please structure your abstract using the PLOS Medicine headings (Background, Methods and Findings, Conclusions).

Please combine the Methods and Findings sections into one section, “Methods and findings”.

Abstract Background: 

Provide expand the details and context of why the study is important – what’s the impact of missed TB cases on health economics, how does it help to identify more cases, for example. 

Abstract Methods and Findings: 

Please ensure that all numbers presented in the abstract are present and identical to numbers presented in the main manuscript text.

Where 95% CIs are reported please also report p-values. Please report exact values or for smaller values, p<0.001

Line 49-51: “Intervention clinics diagnosed 6777 patients with TB [20.7 patients with TB per clinic month (95% CI 16.7, 24.8)] versus 6750 [18.8 patients with TB per clinic month (95% CI 15.3,22.2)].” The use of parentheses is confusing here, especially after p-values are added. Suggest the following:

“Intervention clinics diagnosed 6777 patients with TB, 20.7 patients with TB per clinic month (95% CI [16.7, 24.8], p=/p<…) versus 6750, 18.8 patients with TB per clinic month (95% CI [15.3,22.2], p=/p<…).”

Please amend statistical reporting throughout for consistency and accessibility

Please include the study design (randomised control trial) and further details of the population and setting – what kind of clinics, rural/urban and so on, years during which the study took place, length of follow up, and main outcome measures.

Please include a summary of adverse events if these were assessed in the study.

In the last sentence of the Abstract Methods and Findings section, please describe the main limitation(s) of the study's methodology. 

Abstract Conclusions:

Please replace “discussion with “conclusions” 

Please address the study implications without overreaching what can be concluded from the data; the phrase "In this study, we observed ..." may be useful. What impact does your study have on health/economics/policy change 

Please interpret the study based on the results presented in the abstract, emphasizing what is new without overstating your conclusions.

Please avoid vague statements such as "these results have major implications for policy/clinical care". Mention only specific implications substantiated by the results.

Please avoid assertions of primacy ("We report for the first time....")

Please remove the funding statement from the end of the abstract and include on ly in the submission form

AUTHOR SUMMARY

INTRODUCTION

Please restructure your introduction with the following in mind and in line with reviewer comments below. The editorial team are in agreement with the need to expand the details/justifications for the study design/setting. Please ensure that the introduction addresses past research and explains the need for and potential importance of your study. 

Indicate whether your study is novel and how you determined that. 

If there has been a systematic review of the evidence related to your study (or you have conducted one), please refer to and reference that review and indicate whether it supports the need for your study.

Please conclude the Introduction with a clear description of the study question or hypothesis.

METHODS and RESULTS

Please see statistical reviewer comments regarding your chosen approach to your analyses. Please respond in full

Please see clinical reviewer comments regarding additional details of your study design, sample collection and so on, please revise accordingly. 

Early in the methods section, please indicate the dates during which study enrolment and follow up occurred, the period over which the study conducted and so on

Line 171: Please clarify what is meant by standardised, what standard was set and how it was defined

Line 371: “…(interaction IRR 1.17 95% CI 1.14; 1.19) Please amend as follows: (interaction IRR 1.17 95% CI [1.14, 1.19], p=/p<…) and where 95% CIs are reported please also report p-values (as above)

When a p value is given, please specify the statistical test used to determine it.

Please amend statistical reporting as described above

Please define the length of follow up (e.g, in mean, SD, and range).

The main analysis should be intention to treat (ie, all individuals randomized are included in the analysis in the groups to which they were originally assigned. If the study included dropouts, specify whether their data are imputed and if so using what method. Please refer to as modified ITT). Please state that analysis was intention to treat.

Please provide the number of participants lost to follow up in each group.

Please include the study protocol document and analysis plan, with any amendments, as Supporting Information to be published with the manuscript if accepted.

TABLES and FIGURES

Throughout, please ensure each table/figure has an appropriate caption which clearly describes the table/figure contents without the need to refer to the text.

Please define all any abbreviations in all tables/figures in the caption, including “Xpert MTB/RIF” in table 2, SoC Fig 1S

Please provide a table showing the baseline characteristics of the study population.

Figure 2: please define the numerical values on the y-axis

Figure 3: please also report p-values. Please indicate whether analyses are adjusted and if so which factors are adjusted for. Where adjusted analyses are presented, please also present unadjusted analyses for comparison. 

DISCUSSION

We agree with the reviewers that the discussion should be expanded to address the study design and potential limitations. Please also see reviewer comments below. Please ensure that the discussion is structured as follows

REFERENCES

Please use the "Vancouver" style for reference formatting, and see our website for other reference guidelines here: https://journals.plos.org/plosmedicine/s/submission-guidelines#loc-references

For in-text reference call outs citations should be placed in square brackets and preceding punctuation like so [1,2,3,4] or [1-4,6]. Please amend throughout.

In the bibliography please list up to but no more than 6 author names followed by et al (where more than 6 authors contribute to the study).

Journal name abbreviations should be those found in the National Center for Biotechnology Information (NCBI) databases. 

Comments from the reviewers:

Reviewer #1: Background

* References to the WHO Global TB report can be updated using the 2022 version (now that this is available)

* Line 77-79. While symptom screening does help identify individuals more likely to be diagnosed with TB, it is very subjective and its inconsistent implementation is a big reason for patients being missed with TB. Authors should soften this sentence to rather focus on symptom-based screening being the mainstay for identifying presumptive TB patients and limitations of this approach currently

* Line 79-81. WHO symptom screening has been recommended for persons living with HIV since 2011 and has been shown to be effective - not clear why the authors have suggested otherwise. Is this within the context of South Africa?

* Line 83. Not clear who the 'high risk' groups refer to and why have they been specifically targeted? 

* Line 88-92 should move higher up BEFORE the study hypothesis is presented (line 83)

* The setting of the study is not well captured in the introduction i.e. that you are intending to address undiagnosed TB at clinic-level. In addition, not sufficient justification provided for focusing on contacts of TB patients at facility level - such individuals are actively sought out at their households through contact investigation.

Methods

* Lines 100-102. While indicating the justification for the three provinces is useful, it would also be important to highlight the 'missed/undiagnosed' burden for these provinces as well, since your hypothesis is intended to close this gap?

* Line 103. Confirm which or how many symptoms are considered for further evaluation to be initiated

* Lines 105-107. Please clarify what you mean when you say "Xpert Ultra testing available at virtually all clinics in South Africa"? Are you saying the testing is done at facility level (which is how it currently reads) or that testing is offered at facility-level but samples are sent off to centralized testing labs?

* Lines 111-112. More clarity on how high-risk groups were classified. For PLHIV did it matter if patients were on ART? For contacts of a TB patient, how was this 'contact defined' i.e., household vs casual contact. For previous TB patients, did you determine the outcome of the previous episode?

* Line 123. Why was 15 patients diagnosed per month considered for selecting clinics - how does this align with your hypothesis of 'missed TB patients' that the intervention was going to identify? Was screening and Xpert positivity rates considered at clinic level?

* Lines 135-136. Why was HIV prevalence not considered for your randomization strata - would this not influence yield since you are include PLHIV as one of your high-risk groups?

* Was HIV testing offered as part of the study? It would seem important for individuals in the other two risk groups (contacts and previous TB) to know their status as this would further increase their risk of TB.

* Line 171. Please clarify/provide more information on what 'standardized manner' refers to for sputum collection? How was this standardized?

* Line 174. How were staff trained to assess sputum quality i.e., to ensure that it was mucoid?

* Lines 176-178. When were sputum samples sent, same day or next day to the labs?

* Line 185. Were fieldworks acting on Xpert Ultra results only or also if MGIT cultures were positive?

* Lines 186-188. Please provide more information on whether any of the three provinces considered 'trace' as a positive result?

* Lines 197-201. How were screening and sampling evaluated? Facilities who were notified to the Provincial TB Manager, did they get re-evaluated for improvement? Was there anything done to improve their assessment?

* Line 202. The heading 'Data sources' should be changed to also reflect analyses as the language is focused primarily around describing parts of the analysis. I would suggest include a definitions sub-heading to clarify who was deemed a TB patient. Could also include other classifications as well such as the high-risk groups.

* Line 221. Please provide justification for the assumed 25% difference in yield between intervention and control clinics.

* Line 243-244. For individuals with an initial 'trace' result, were these patients evaluated further? In intervention clinics, MGIT culture was performed in addition to Xpert Ultra and thus patients in this arm who had an initial trace result would have benefited immediately from the additional culture investigation, which may or may not have been done for trace patients in the control arm (as they would have also needed to provide an additional specimen). Would this have biased the outcome of your comparison?

Discussion

* Line 392. Remove 'statistically' to make the sentence clearer.

* Line 401-403. Cost-effectiveness would be an important consideration for this strategy and should be outlined as a follow-up to this manuscript. While the intervention increased TB detection by 17% among intervention clinics, it does not reflect the resources required to achieve this increase. In the absence of cost-effectiveness data, would it be possible for the authors to comment on the number-needed-to-test (NNT) in intervention clinics among enrolled participants; this could provide some perspective on the effort/resources required?

* Line 408-411. While it is possible that TUTT may help increase testing at clinic level, their needs to be a shift towards community interventions and increased case finding (especially individuals who are not presenting to health facilities). Is there a role for TUTT in community interventions (considering the same high risk groups, particularly household contacts of TB patients? Please provide additional commentary around this.

* Discussion requires more synthesis of important aspects relevant to the TUTT intervention. Firstly, an inherent limitation of TUTT is sputum collection since individuals without symptoms at the time of investigation are less likely to provide a quality sputum. Can results be presented around sputum quality, for intervention participants and how this may or may not have affected the study. In addition, there is no mention of the potential of urine LAM (and its current use in SA) or other alternative sample types, particularly since the former is currently aimed at PLHIV. Secondly, there is a lack of discussion on how whether the choice of risk groups was the correct groups to have 'targeted' for the intervention. PLHIV are routinely screened for TB so unclear if the intervention would have supplemented such efforts within facilities. Not being able to disaggregate the risk group data from the overall data is a limitation to the study as it doesn't address the question about whether the correct groups were selected. This requires some discussion. Thirdly, and related to the previous point, only adults were considered for the study, and while it makes it easier re. informed consent and sputum collection, there is a desperate need to improve TB detection in children, particularly the under 5 year age group. The authors should provide some discussion around this and how TUTT could evolve (with newer tools and sample types) to address these other equally important populations. Fourthly, the use of culture as part of the investigation of deviates from current practice within the programme. This needs to be included more clearly as a limitation, however does require further discussion, particularly in relation to trace results. 

* Related to cost-effectiveness, was this included as part of the study? 

* Digital chest x-ray has shown to be effective at identifying individuals at primary healthcare level and reducing the NNT by half compared to symptom screening (Moodley et al. CID. 2022; PMID: 34313729). Some discussion and reference is needed around the utility of chest xray for identifying undiagnosed TB and whether such an intervention is more scalable for programmes compared to universal testing.

* As this is a RCT, a CONSORT checklist should be included

Reviewer #2: A Cluster Randomized Trial of Systematic Targeted Universal Testing for Tuberculosis

in Primary Care Clinics of South Africa

PMEDICINE-D-22-03274R1

Thank you for asking me to review this manuscript which goes some way to identifying more patients with TB compared to the current standard of care in South Africa. In this cluster randomised trial, the authors evaluate whether targeted universal testing for TB (TUTT) in high risk groups identified more patients with TB compared to the routine standard of care. The dogged determination and resilience of the study team in completing the study despite delays in approvals and clinics burning down etc needs to be applauded. Consequently, we have study results from the real world which can inform policy. 

I have no major comments or concerns with this well-written, detailed manuscript, only one question and a couple of minor editing issues. However, I am not a statistician so cannot comment on the statistical analysis conducted.

Supplementary Figures 1a and b: In December 2019 in the control clinics the number of TB cases decreased by 5 or 6. What is the reason for the decrease in over 10 TB cases in December 2019 in the intervention clinics? 

Lines 223 - 224: 'We wanted to obtain a full calendar year of data from each clinic to account for seasonal changes.' This has already been stated in line 114 and could be deleted.

Line 268. Spelling of additionally.

Lines 341 - 342: '…..the coefficient of variation in of the number of diagnoses in the control arm was 0.4.' This sentence is confusing and needs revising.

Reviewer #3: This cluster randomized control trial tried to evaluate whether targeted universal testing for TB (TUTT) is more effective in identifying TB cases than the standard of care (i.e., systematic symptom screening for TB) in South Africa. Overall, this is a well-designed and conducted study. The topic of the study is of significant importance and the results are relevant for policy makers. Below are my specific comments.

1. The authors' institute received funding from Pfizer, which may result in potential competing interest.

2. The date are not fully public available which is not aligned with PLoS Medicine's policy. 

3. Line 149-151: Since there will be a fieldworker allocated to be present at each intervention clinic but not at control clinic, is that really possible to achieve masking for this study?

4. Line 157-163: It said (line 157-160) that attendees who thought they were in a high risk group to privately contact the fieldworker. But in lines 162-163, it says that patients were referred by the healthcare workers. It is a bit confusing to me. By which way are the patients being identified?

5. Line 166-168: why? What if the last time test was negative. Should the patients still be eligible?

6. Line 235-238: A CV of 0.24 or 0.34? You mentioned in line 230-231 that "the observed between-facility coefficient of variance (CV) was 0.34". Where did the 0.24 come from? 

7. Line 253-258: I think using negative binomial model may be better than using Poisson regression due to the restrictive assumption about mean and variance (mean=variance) for Poisson regression.

[LINK]

---

## [Decision Letter · Decision Letter 2]

3 Mar 2023

Dear Dr. Martinson,

Thank you very much for submitting your manuscript "A cluster randomized trial of systematic targeted universal testing for tuberculosis in primary care clinics of South Africa (The TUTT Study)" (PMEDICINE-D-22-03274R2) for consideration at PLOS Medicine. 

[LINK]

Following discussion, I am afraid that we will not be able to accept the manuscript for publication in the journal in its current form, but we would like to consider a revised version that addresses the editors' comments. Obviously we cannot make any decision about publication until we have seen the revised manuscript and your response, we may need to seek re-review by one or more of the reviewers. 

In revising the manuscript for further consideration, your revisions should address the specific points made by each the editors. Please also check the guidelines for revised papers at http://journals.plos.org/plosmedicine/s/revising-your-manuscript for any that apply to your paper. In your rebuttal letter you should indicate your response to the editors' comments, the changes you have made in the manuscript, and include either an excerpt of the revised text or the location (eg: page and line number) where each change can be found. Please submit a clean version of the paper as the main article file; a version with changes marked should be uploaded as a marked up manuscript.

We expect to receive your revised manuscript by Mar 24 2023 11:59PM. Please email us (plosmedicine@plos.org) if you have any questions or concerns.

We look forward to receiving your revised manuscript. 

Sincerely,

Philippa Dodd, MBBS MRCP PhD

PLOS Medicine

plosmedicine.org

GENERAL

Thank you for your detailed and considerate responses to previous editor and reviewer requests, please see below for additional comments which we require that you address in full.

** We note in your abstract that the recruitment to the study was “halted” in March 2020 (coinciding with pandemic lockdowns). Please include details, in the abstract (and the main manuscript) which clearly states why the trial was halted and whether termination of recruitment at this time point has any implications on sample size and statistical powering (you go some way to doing this in the discussion) **

DATA AVAILABILITY

In accordance with ICMJE requirements, PLOS Medicine requires prospective, public registration of a data sharing plan (as part of mandatory clinical trials registration) for all clinical trials that began enrolment on or after January 1, 2019. Please include details.

TITLE

Please revise your title according to PLOS Medicine's style. Your title must be nondeclarative and not a question. It should begin with main concept if possible. "Effect of" should be used only if causality can be inferred, i.e., for an RCT. Please place the study design ("A randomized controlled trial," "A retrospective study," "A modelling study," etc.) in the subtitle (ie, after a colon).

ABSTRACT

Line 52: “Subjecting data extracted…” instead perhaps?

Line 50: please revise to read “…attendees living with HIV…” please check and amend throughout the manuscript where relevant.

Please include the primary outcome.

AUTHOR SUMMARY

Thank you for including an author summary. Please trim your summary to include ideally 2-3 (but no more than 4) single sentence bullet points for each of the questions (Why Was This Study Done?, What Did the Researchers Do and Find? , What Do These Findings Mean?). Bullet points should be objective, brief, succinct, specific, accurate, and avoid technical language.

Reviewing existing published articles here https://journals.plos.org/plosmedicine/ on our website may be helpful to you.

INTRODUCTION

Line 178: as above please amend to read as follows, “…in people living with HIV…”. Please check and amend throughout the manuscript where relevant.

METHODS and RESULTS

PLOS Medicine requires that all trials be prospectively registered in one of registries recognized by WHO. Please provide information on study registration in the Methods section.

Please amend statistical reporting for consistency. For example, line 450 reads, “…1.14 (95% CI 0.94, 1.38, p=0.46)” but at line 457, “…1.19 (0.99, 1.44) (p=0.05).” Please revise throughout for consistency. We suggest adopting the former style of presentation.

Line 269: Please include whether consent obtained was written or oral.

PARTICIPANT CHARACTERISTICS

Thank you for including details of the clinics include in your trial.

Throughout, specific individual patient characteristics are documented to be required for inclusion into the study, including “high risk” status and the reasons for the risk. Further you specifically document individual consent to participate in the study. For these reasons and to help facilitate transparency of data reporting we request that you please include the characteristics of the consenting participants.

FIGURES

Figure 3 – when reporting p values, please include numerical values instead of using asterisks as indicators

STATISTICAL ANALYSIS PLAN

In your rebuttal, you state that no statistical analysis plan was made but in the abstract, you write “pre-specified difference in-differences analyses” and in the manuscript submission form you state “the local IRB has acknowledged both the planned analysis and…” which would suggest you had a pre-specified analysis plan please clarify/amend to include the statistical analysis plan that you refer to.

DISCUSSION

Please remove the declarations of interest statement and the data sharing statements from the end of the discussion and include only in the manuscript submission form.

REFERENCES

For in-text reference callouts, please ensure a) an absence of spaces between citations and b) a space preceding the opening parenthesis. For example, line 147 “…cases (10 million) [1,2]…”. Please check and amend throughout.

Comments from the reviewers:

Reviewer #1: My comments have been satisfactorily addressed.

Reviewer #3: I think the authors have adequately answered/addressed my questions and questions from the other reviewers. I think the paper is acceptable in its current form. Congratulations!

[LINK]

---

## [Editor Report · Decision Letter 3]

6 Apr 2023

Dear Dr. Martinson,

Thank you very much for re-submitting your manuscript "Evaluating systematic targeted universal testing for tuberculosis in primary care clinics of South Africa; a cluster randomized trial (The TUTT Trial)" (PMEDICINE-D-22-03274R3) for review by PLOS Medicine.

I have discussed the paper with my colleagues and I am pleased to say that provided the remaining editorial and production issues are dealt with we are planning to accept the paper for publication in the journal.

[LINK]

We look forward to receiving the revised manuscript by Apr 13 2023 11:59PM.   

Sincerely,

Philippa Dodd, MBBS MRCP PhD

Senior Editor 

PLOS Medicine

plosmedicine.org

Requests from Editors:

GENERAL

Thank you for your detailed responses to previous editor and reviewer comments, please see below for further comments which we require that you address prior to publication.

AUTHOR SUMMARY

Thank you for revising the author summary. Please see below for recommended revisions, as per our understanding, to improve accessibility and brevity. Please check carefully to ensure accuracy. Where [….] are presented please fill in the blanks

Why Was This Study Done? 

o TB diagnosis is made by laboratory sputum testing usually prompted by the presence of at least one symptom of TB (cough, fever or night sweats, loss of weight) 

o The World Health Organization promotes screening of entire groups at high risk of TB irrespective of symptoms. In South Africa high risk groups include people living with HIV; those who report being a close contact of someone with TB; and those who have had a recent episode of TB.

o A sparsity of data exists on the usefulness of diagnostic assays as screening tests. Our trial assessed if targeted testing of people at high risk without TB symptoms helped to identify undiagnosed TB.

What did the researchers do and find?

o 62 large primary care clinics in South Africa were randomised to either our intervention -TB testing of all people in high-risk groups, or our control - to continue symptom based testing for TB (the standard of care). 

o The total number of TB cases diagnosed each month in all clinics was recorded. Comparisons were made between intervention and control clinics. 

o In the intervention clinics 6777 people were diagnosed with TB, an average of 20.7 patients per clinic month versus 6750 in control clinics, an average of 18.8 patients with TB per clinic month. 

o After adjusting for […….] intervention clinics were observed to diagnose 14% more patients with TB than control clinics but this did not reach statistical significance. Secondary analyses, including data from the year prior to the intervention demonstrated a statistically significant increase in TB diagnoses per month, reported as a 17% relative increase.

What do these findings mean?

o A strategy targeting high risk groups for universal testing for TB (TUTT) may help to improve diagnostic rates of TB in areas where prevalence is high.

o Less expensive TB tests could enable universal testing strategies to be implemented in low-resource settings; and could be extended to other high risk target groups.

PARTICIPANT CHARACTERISTICS

We previously said the following:

“Thank you for including details of the clinics include in your trial.

Throughout, specific individual patient characteristics are documented to be required for inclusion into the study, including “high risk” status and the reasons for the risk. Further you specifically document individual consent to participate in the study. For these reasons and to help facilitate transparency of data reporting we request that you please include the characteristics of the consenting participants.”

Author response: “We have included a brief summary but in the text refer the reader to this trial’s related manuscript recently published in CID that has a detailed description of the individual participants we consented and recruited but not the trial outcomes. The CID paper should be read in conjunction with the manuscript under review. Please see table 1 in our CID paper: https://academic.oup.com/cid/advancearticle/doi/10.1093/cid/ciac965/6969441?login=false.”

We were unfortunately unable to access the manuscript via the link above but the editorial team agree that it would be unfair to expect the reader to refer to another paper for this information.

Please include a table of baseline characteristics of your study population within either the main manuscript or as supporting information. Please ensure that in the manuscript text you direct the reader to where this can be found. This is a prerequisite for publication.

FIGURE 3

The formatting of this figure is inconsistent with PLOS Medicine’s formatting requirements and requires revision.

Numerical values should be present as 0.943, as opposed to .943. Please revise.

Please ensure that the solid dividing lines of your plots do not strike through the plot titles (fig 3b). Titles/headers should be placed above the plots.

Asterisks are not permitted please revise to include p values. Please report as p<0.001 and where higher as p=0.002, for example. Please do not report as p<.001 or as p=.002

In figure 3a there are both asterisks and complete values (2nd row ‘medium’). At line 4 (‘GP’) dots and lines overlap numerical values and text – please revise.

Please ensure that all abbreviations are defined (IRR, CI, GP, KZN, WC) either in an appropriate footnote or in the figure caption.

REFERENCES

For all web references please include an access date

CONSORT CHECKLIST

Please refer to section and paragraph numbers as opposed to page/line numbers as these often change at the time of publication.

SOCIAL MEDIA

To help us extend the reach of your research, if not already done so, please provide any Twitter handle(s) that would be appropriate to tag, including your own, your co-authors’, your institution, funder, or lab. Please detail any handles you wish to be included when we tweet this paper, in the manuscript submission form when you re-submit the manuscript.

[LINK]

---

## [Editor Report · Decision Letter 4]

21 Apr 2023

Dear Dr Martinson, 

On behalf of my colleagues and the Academic Editor, Dr. Amitabh Suthar, I am pleased to inform you that we have agreed to publish your manuscript "Evaluating systematic targeted universal testing for tuberculosis in primary care clinics of South Africa; a cluster randomized trial (The TUTT Trial)" (PMEDICINE-D-22-03274R4) in PLOS Medicine.

PRESS

Best wishes,

Pippa 

Philippa Dodd, MBBS MRCP PhD 

PLOS Medicine